# Propranolol Modulates Cerebellar Circuit Activity and Reduces Tremor

**DOI:** 10.3390/cells11233889

**Published:** 2022-12-01

**Authors:** Joy Zhou, Meike E. Van der Heijden, Luis E. Salazar Leon, Tao Lin, Lauren N. Miterko, Dominic J. Kizek, Ross M. Perez, Matea Pavešković, Amanda M. Brown, Roy V. Sillitoe

**Affiliations:** 1Department of Pathology and Immunology, Baylor College of Medicine, Houston, TX 77030, USA; 2Jan and Dan Duncan Neurological Research Institute, Texas Children’s Hospital, 1250 Moursund Street, Suite 1325, Houston, TX 77030, USA; 3Department of Neuroscience, Baylor College of Medicine, Houston, TX 77030, USA; 4Program in Development, Disease Models & Therapeutics, Baylor College of Medicine, Houston, TX 77030, USA; 5Department of Pediatrics, Baylor College of Medicine, Houston, TX 77030, USA

**Keywords:** cerebellum, tremor, propranolol, circuitry, beta-adrenergic receptors, electrophysiology

## Abstract

Tremor is the most common movement disorder. Several drugs reduce tremor severity, but no cures are available. Propranolol, a β-adrenergic receptor blocker, is the leading treatment for tremor. However, the in vivo circuit mechanisms by which propranolol decreases tremor remain unclear. Here, we test whether propranolol modulates activity in the cerebellum, a key node in the tremor network. We investigated the effects of propranolol in healthy control mice and *Car8^wdl/wdl^* mice, which exhibit pathophysiological tremor and ataxia due to cerebellar dysfunction. Propranolol reduced physiological tremor in control mice and reduced pathophysiological tremor in *Car8^wdl/wdl^* mice to control levels. Open field and footprinting assays showed that propranolol did not correct ataxia in *Car8^wdl/wdl^* mice. In vivo recordings in awake mice revealed that propranolol modulates the spiking activity of control and *Car8^wdl/wdl^* Purkinje cells. Recordings in cerebellar nuclei neurons, the targets of Purkinje cells, also revealed altered activity in propranolol-treated control and *Car8^wdl/wdl^* mice. Next, we tested whether propranolol reduces tremor through β_1_ and β_2_ adrenergic receptors. Propranolol did not change tremor amplitude or cerebellar nuclei activity in β_1_ and β_2_ null mice or *Car8^wdl/wdl^* mice lacking β_1_ and β_2_ receptor function. These data show that propranolol can modulate cerebellar circuit activity through β-adrenergic receptors and may contribute to tremor therapeutics.

## 1. Introduction

Tremor is the most common movement disorder, affecting an estimated 70 million people worldwide [1]. The main symptoms involve uncontrollable oscillations of the limbs and head. When severe, these altered movements decrease a patient’s quality of life [2]. The etiology of tremor is diverse, ranging from genetic and environmental factors [3,4,5,6] to arising as a side effect from medications or comorbidity with other diseases such as multiple sclerosis, diabetes, and neuropsychiatric disorders [7,8,9]. There are several characteristics that define tremor, although oscillatory frequency is the primary feature that determines its clinical diagnoses. Essential tremor, which is the most common tremor-related diagnosis worldwide [1], typically occurs at a frequency between 8 and 12 Hz. Essential tremor amplitude can increase as the disease progresses [10,11,12] or transiently decrease with alcohol consumption [13]. Essential tremor can also be comorbid with other neurological conditions that involve the cerebellum such as ataxia and dystonia [14,15,16], and also with other insults that impact the functional anatomy of the cerebellum [17,18,19,20,21].

Despite its prevalence, the mechanisms that initiate tremor within the central nervous system remain poorly understood, and as a result, treatment options are limited. For severe cases of essential tremor, surgeries including thalamotomy or deep brain stimulation (DBS) are often the only effective ways to relieve symptoms [22,23,24,25]. As such, DBS has become the main therapeutic consideration for patients with intractable tremor [26]. Focused ultrasound thalamotomy offers a non-invasive method to treat tremor, although patients must consider the ablation of the targeted region [27,28,29]. However, prescription drugs are the predominant treatment for most essential tremor patients. β-blockers, which include propranolol and metoprolol, are the most common treatments for essential tremor. Propranolol is effective at reducing tremor severity in 50–70% of patients [30,31,32,33].

Propranolol is a non-selective β-blocker that counters the action of adrenaline and noradrenaline at both β_1_ and β_2_ adrenergic receptors throughout the body [34,35]. Propranolol is used to treat a variety of disorders, ranging from hypertension and anxiety to migraines and tachycardia. Its efficacy for tremor was first demonstrated in 1965 [36], and it has been widely prescribed for this condition since 1967 [37]. Propranolol crosses the blood-brain barrier, and as a strongly lipophilic drug it reaches high concentrations in the brain [38,39,40,41]. Despite being the leading treatment for tremor, its mechanism of action for reducing tremor severity is still unknown. Prior studies have suggested that propranolol may act upon β-adrenergic receptors in the peripheral nervous system and muscle spindles during tremor attenuation [42,43,44,45,46,47]. Though among these previous efforts, only a few mechanistic in vivo studies considered the effects of propranolol in the central nervous system, with suggestions that it may provide a sedative effect [48,49,50,51]; however, the mechanisms of propranolol on tremor in the central nervous system remain unknown. Furthermore, it is now accepted that tremor can be generated from within the central nervous system [52], raising the possibility that propranolol has critical targets in the brain.

The cerebello-thalamo-cortical pathway is likely a key circuit for mediating tremor [53,54,55], although the role that each node has in the circuit is unclear [56]. Evidence points to the cerebellum as a potential inception site for tremor, with studies showing that disrupted function of the Purkinje cell microcircuit (with the upstream inferior olive and downstream cerebellar nuclei) contributes to the pathophysiology [18,57]. Postmortem studies of human patients with essential tremor have also shown anatomical abnormalities in the cerebellar cortex [58]. Consistent with these data, it is thought that cerebellar circuits are sufficient to drive tremor [59,60]. Additionally, cerebellar-targeted DBS is effective in resolving tremor in mice [60] and human patients [61,62].

Here, we investigated whether propranolol reduces tremor through changes in cerebellar circuit activity, by testing whether propranolol’s action is mediated through β_1_ and β_2_ adrenergic receptors. We used *Car8^wdl/wdl^* mutant mice to test how cerebellar dysfunction responds to propranolol in a genetic mouse model with severe tremor. Importantly, the pathology and dysfunction of *Car8^wdl/wdl^* mice involves a functional defect within the cerebellum, but is free from gross anatomical disruptions [63]. Mutations in the carbonic anhydrase VIII (*Car8*) gene results in loss of the CAR8 protein. CAR8 expression is initiated during development and is predominantly expressed in cerebellar Purkinje cells. *Car8^wdl/wdl^* mice have an action tremor with an amplitude that worsens with age, peaks at ~8–12 Hz [63], and is suppressed with alcohol [64]. The tremor in *Car8^wdl/wdl^* mice is accompanied by ataxia and dystonia [63,65,66]. We used a combination of propranolol as a pharmacological tool, tremor severity analysis, in vivo electrophysiology, and β-adrenergic receptor mutant mice to identify specific cerebellar substrates of tremor reduction in control and *Car8^wdl/wdl^* mice. Our studies aim to uncover how propranolol operates at the cellular and circuit levels in the cerebellum while reducing tremor symptoms.

## 2. Materials and Methods

### 2.1. Animal Maintenance

Mouse husbandry and experiments were performed under an approved Institutional Animal Care and Use Committee (IACUC) protocol at Baylor College of Medicine (BCM). Male and female mouse genetic models (see the details below for the different alleles) were obtained from The Jackson Laboratory (Bar Harbor, ME, USA) and a colony was established and thereafter maintained at BCM. Mice of both sexes were studied. All mice used in this study were mature adults, with their ages ranging between 4 and 12 months old. Detailed sample sizes of animals and data points of all mouse genotypes and conditions used in this study are described in the figure legends and Appendix A.

### 2.2. Genetically Engineered Mouse Lines

C57BLKS/J controls, *Car8^wdl/wdl^* mutants, and *β1-AR^−/−^;β2-AR^−/−^* mutants were purchased from The Jackson Laboratory (Bar Harbor, ME, USA). The waddles mutation in the *Car8^wdl/wdl^* mutants arose spontaneously in 1995 and has been maintained by The Jackson Laboratory since. For *β1-AR^−/−^;β2-AR^−/−^* double mutant mice, the original strain was created as described by Rohrer et al., 1999 [67], by mating *β1-AR^−/−^* homozygous mutant mice with *β2-AR^−/−^* homozygous mutant mice to generate compound heterozygotes, the offspring of which were then mated to obtain compound homozygotes. *Β1-AR^−/−^* mice were created using a targeting vector containing a neomycin resistance gene driven by the mouse phosphoglycerate kinase promoter to disrupt most of the Adrb1 coding region (all but a 3′ 153 bp segment). *β2-AR^−/−^* mutant mice were created in a similar fashion using a targeting vector, again containing a neomycin resistance gene driven by the mouse phosphoglycerate kinase promoter to disrupt Adrb2 such that the end of the fourth transmembrane segment is absent, rendering the receptors nonfunctional. To generate *Car8^wdl/wdl^* mice lacking β-adrenergic receptors, *Car8^wdl/wdl^* mice were crossed with *β1-AR^−/−^;β2-AR^−/−^* mice to generate *β1-AR^−/−^;β2-AR^−/−^;Car8^wdl/wdl^* triple allele mutants. We used a standard PCR genotyping protocol to differentiate the mutants from the controls, using the primer sequences listed in Appendix A. All mice were maintained in our animal colony under an approved IACUC animal protocol according to the institutional guidelines at BCM.

### 2.3. Propranolol Administration

Adult mice were administered a 20 mg/kg propranolol hydrochloride (Sigma-Aldrich, St. Louis, MO, USA; P0884-1G) solution dissolved in sterilized distilled water via intraperitoneal injection (IP) with a 31-gauge 6 mm syringe. A total of 20 mg/kg is a clinically moderate dosage for human patients on propranolol treatment for tremor. A reduction in tremor consistently developed between 10 and 20 min post-injection, and data recordings were made no shorter than 15 min and no longer than 6 h after injection. Mice were given one dose of propranolol, one time only, to avoid any potential cumulative effects.

### 2.4. Tremor Recording and Analysis

Tremor was detected using a custom-built tremor monitor. For each experiment, mice are placed into a translucent plastic box with an open top. The box is held firmly in midair by eight elastic cords. An accelerometer is attached to the bottom, allowing for signal detection of oscillatory movements in two dimensions. Further details can be found in our previous publication [68].

Mice were allowed to acclimate to the tremor monitor for 300 s before recordings of tremor were made. All tremor monitor recordings were sampled at 10 kHz and lowpass filtered at 5 kHz. Power spectrums of tremor were made using a fast Fourier transform (FFT) with a Hanning window. An offset was applied if the tremor waveform was not centered on zero and the recordings were downsampled using the Spike2 software ‘interpolate’ channel process in order to produce frequency bins aligned to whole numbers. FFT frequency resolution was targeted to either ~0.25 Hz or ~0.5 Hz per bin. Sonogram plots of tremor were made using the Spike2 software ‘sonogram’ channel draw mode with a Hanning window. We report frequency ranges in order to account for natural variability in the tremor signals. The entirety of each 300 s post-acclimation tremor recording was analyzed for each animal in each group.

### 2.5. Non-Tremor Movement Events Detection

To examine whether non-tremor movements may affect overall tremor readings, non-tremor movement information was extracted from each tremor recording by applying a power spectrum analysis using an FFT to the raw tremor monitor data to determine the power within the 50–100 Hz range. An offset was applied to the raw waveform before analysis if it was not centered on zero. FFT frequency resolution was targeted to 19.54 Hz with a window size of 0.0512 s. Because clinical tremors are categorized at frequencies below 30 Hz, the 50–100 Hz range of data collected by the tremor monitor represents non-tremor movements. Furthermore, a detection threshold of 50 mV^2^ of power was set so that only strong, sudden movements–such as footsteps and rearing–would be captured as non-tremor movements. Each event surpassing the detection threshold occurring at least 0.1 s apart from the prior or following event was counted as a single movement, and the total number of such detected movements over the entire 300 s tremor monitor recording was analyzed for each animal in each condition.

### 2.6. Open Field Assay

General locomotor function was assessed with the open field assay using the open field locomotion system (Omnitech Electronics, Inc., Columbus, OH, USA), as described in Miterko et al., 2021 [69]. For both the control and *Car8^wdl/wdl^* baseline (no propranolol) conditions, the data analyzed and represented in this study are taken from the database used in Miterko et al., 2021, from which we extracted data for control and *Car8^wdl/wdl^* mice at baseline to make the direct comparison between the two genotypes by pooling baseline data across different “frequency” groups published in Miterko et al., 2021. For the propranolol-treated conditions of control and *Car8^wdl/wdl^* mice, new data was collected; since mice quickly acclimate to the open field assay, the same mouse cannot be recorded from twice. Baseline and propranolol-treated data are, therefore, from two separate cohorts of mice of each genotype. Mice were treated with propranolol, then acclimated to the open field room for 1.5 h. Immediately following the acclimation period, mice were placed in the open field apparatus, where their movement was recorded for 30 min. White noise was provided in the background during the acclimation and the recording periods. The movement time, number of movement events, and number of ambulatory events of each mouse were analyzed (Omnitech Electronics, Columbus, OH, USA; Accuscan Fusion Software, Version 4.7).

### 2.7. Footprinting Assay

The gait characteristics of control, *Car8^wdl/wdl^*, and *β1-AR^−/−^;β2-AR^−/−^* mice before and after propranolol treatment were assessed using the footprinting assay as described in Carter et al., 1999 [70]. To track footprints, the paws of the mice were coated with nontoxic brightly colored paint. The animals were then allowed to walk along a 50 cm-long, 10 cm-wide tunnel (with 10 cm-high walls) into a light-shielded, enclosed space. All mice had three runs each, both before and after propranolol treatment. A fresh sheet of white paper was placed on the floor of the tunnel for each run. Three step parameters were recorded from each footprinting sheet (all measured in centimeters). Stride length was measured as the distance of forward movement between each paw for three steps. Sway distance was measured by recording the perpendicular distance of a given paw to a line connecting its opposite paw’s preceding and proceeding steps. Stance was measured as the hypotenuse of stride and sway lines. For each step parameter, three steps from each paw were measured from each run, excluding footprints made at the beginning and end of the run where the animal was initiating and finishing movement, respectively. Only forepaw measurements are reported and represented in our data visualizations. The mean value of each set of three values was used in subsequent analyses.

### 2.8. Surgery for Awake Head-Fixed Neural Recordings

We have previously described our general surgical techniques in detail [71]. In brief, for all surgical techniques used in these studies, mice were given preemptive analgesics (slow-release buprenorphine, 1 mg/kg subcutaneous (SC), and meloxicam, 5 mg/kg SC) with continued applications provided as part of the post-operative procedures. Anesthesia was induced with 3% isoflurane gas and maintained during surgery at 2% concentration. All surgeries were performed on a stereotaxic platform (David Kopf Instruments, Tujenga, CA, USA) with sterile surgery techniques. All animals were allowed to recover for at least three days to a maximum of one week after surgery.

During surgeries for awake neural recording experiments, the dorsal aspect of the skull was exposed, and a circular craniotomy of about 2 mm in diameter was performed dorsal to the interposed nucleus (6.4 mm posterior and ±1.3 mm lateral to bregma). A custom-built 3D-printed chamber was placed around the craniotomy and filled with antibiotic ointment. A custom headplate used to stabilize the mouse’s head during recordings was affixed over bregma, and a skull screw was secured into an unused region of the skull. All implanted items were secured using C and B Metabond Adhesive Luting Cement (Parkell, Edgewood, NY, USA) followed by a layer of dental cement (A-M Systems, Sequim, WA, USA; dental cement powder #525000 and solvent #526000) to completely enclose the area.

### 2.9. In Vivo Electrophysiology

Single-unit extracellular recordings were performed as previously described [60,71,72,73,74]. The mice were awake and head-fixed to a frame while standing on a foam wheel, which reduced the force they were able to apply to the headplate. Before recordings, the mice were trained to become accustomed to being in the recording setup and head-fixed. At the time of the recordings, the chamber around the craniotomy was emptied of antibiotic ointment and refilled with 0.9% *w*/*v* NaCl solution. Electrodes had an impedance of 4–13 MΩ and were made of tungsten (Thomas Recording, Giessen, Germany). Electrodes were connected to a preamplifier headstage (NPI Electronic Instruments, Tamm, Germany). The headstage was attached to a motorized micromanipulator (MP-225; Sutter Instrument Co., Novato, CA, USA). Headstage output was amplified and bandpass filtered at 0.3–13 kHz (ELC-03XS amplifier, NPI Electronic Instruments, Tamm, Germany) before being digitized (CED Power 1401, CED, Cambridge, UK), recorded, and analyzed using Spike2 software (CED, Cambridge, UK). Electrical activity was additionally monitored aurally using an audio monitor (AM10, Grass Technologies, West Warwick, RI, USA) connected to the output of the amplifier.

Purkinje cells were identified by their firing rate and the presence of both complex and simple spike activity. Accordingly, cerebellar nuclei cells were identified by their relatively deep location within the cerebellum, approximately 2.5–3 mm deep, and their firing rate. The surface of the tissue was determined based on the significant reduction in electrical noise that occurs when the electrode, initially suspended in air, touches the tissue. During recording sessions, the experimenter monitored the sound of the electrical activity to determine whether tissue membranes were being breached, white matter tracts were being traversed, or whether sound quality deviated significantly from traditional cerebellar recordings. Using these criteria, we have consistently been able to target the cerebellar nuclei [75]. Only traces with clearly identifiable complex spikes were included for Purkinje cells and, for all cells, only those with sufficiently long and stable recording times (113 ± 7 s average; minimum = 40 s) with an optimal signal-to-noise ratio were included [76]. The distribution of cells recorded from each genotype in each condition is described in the figure legends.

Firing properties were analyzed using Spike2 (CED, Cambridge, UK, version 7.2.0), Microsoft Excel (Microsoft, Redmond, WA, USA; 2022 version), custom MATLAB code (MathWorks Inc., Natick, MA, USA; version R0222a), and GraphPad Prism (GraphPad Software, La Jolla, CA, USA, version 9.4.1) software. All electrophysiological recording data were spike sorted in Spike2. We sorted out three types of spikes: Purkinje cell simple spikes, Purkinje cell complex spikes, and cerebellar nuclei spikes. Complex spikes were characterized by their large amplitude, and post-spike depolarization and the smaller spikelets that follow. All other Purkinje cell action potentials were characterized as simple spikes, and all spikes from cerebellar nuclei cells were characterized as such. After spike sorting our traces in Spike2, we calculated several parameters to describe the firing rate and regularity in MATLAB (MathWorks Inc., Natick, MA, USA; version R2022a) as described in Van der Heijden et al., 2021 [77]. For this study, we defined ‘firing rate’ as the number of all spikes observed in the total analyzed recording time (spikes/s). Mode ISI^−1^ was calculated by taking the inverse of the interspike intervals (ISI^−1^) and distributing these in 10 Hz bins. We then found the bin with the largest proportion of spikes and defined the mode ISI^−1^ as the center of that bin (5, 15, 25, etc.). Our measures of global regularity or burstiness (CV) and regularity (CV2) were based on the interspike intervals (ISI) between two adjacent spikes (in seconds). CV = standard deviation(ISI)/mean(ISI), and CV2 = mean(2*|ISI_n_ − ISI_n−1_|/(ISI_n_ + ISI_n−1_)) [78].

### 2.10. Immunohistochemistry

Perfusion and tissue fixation were performed as previously described [68]. Briefly, mice were anesthetized by intraperitoneal injection with Avertin (2, 2, 2-Tribromoethanol, Sigma-Aldrich, St. Louis, MO, USA; catalog # T4). Cardiac perfusion was performed with 0.1 M phosphate-buffered saline (PBS; pH 7.4), then by 4% paraformaldehyde (4% PFA) diluted in PBS. For cryoembedding, brains were post-fixed at 4 °C for 24 to 48 h in 4% PFA and then cryoprotected stepwise in sucrose solutions (15 and 30% diluted in PBS) and embedded in Tissue-Tek O.C.T. compound (Sakura Finetek, Torrance, CA, USA; catalog #4583). Tissue sections were cut on a cryostat with a thickness of 40 μm and individual free-floating sections were collected sequentially and immediately placed into PBS. Our procedures for immunohistochemistry on paraffin and free-floating frozen cut tissue sections have been described extensively in previous work [72,75,79,80,81]. After completing the staining steps, the tissue sections were placed on electrostatically coated glass slides and allowed to dry.

Rabbit polyclonal anti-β_1_ receptor (1:200; Invitrogen, Waltham, MA, USA; catalog #PA1-049, RRID:AB_2289444) and anti-β_2_ receptor (1:200; abcam; Cambridge, UK; catalog#ab182136, RRID:AB_2747383) antibodies were used to label both adrenergic receptor subtypes in the cerebellum. Mouse monoclonal anti-calbindin D28 (1:10,000; Swant, Burgdorf, Switzerland; catalog #300) was used to label the morphology of adult Purkinje cells. Calbindin D28 encodes a calcium binding protein that is expressed exclusively within Purkinje cells of the cerebellum [82]. We visualized immunoreactive complexes either using diaminobenzidine (DAB; 0.5 mg/mL; Sigma-Aldrich, St. Louis, MO, USA) or fluorescent secondary antibodies. For the DAB reaction, we used horseradish peroxidase (HRP)-conjugated goat anti-rabbit and goat anti-mouse secondary antibodies (diluted 1:200 in PBS; Agilent DAKO, Santa Clara, CA, USA) to bind the primary antibodies. Antibody binding was revealed by incubating the tissue in the peroxidase substrate 3,3′-diaminobenzidine tetrahydrochloride (DAB; Sigma-Aldrich, St. Louis, MO, USA; catalog #D5905), which was made by dissolving a 100 mg DAB tablet in 40 mL PBS and 10 μL 30% H_2_O_2_. The DAB reaction was stopped with PBS when the optimal color intensity was reached. Staining for fluorescent immunohistochemistry was performed using donkey anti-mouse or anti-rabbit antibodies conjugated to Alexa-488 and -555 fluorophores (1:1500 for both; Invitrogen, Waltham, MA, USA). Tissue sections were coverslipped using either Entellan mounting media (for DAB; Electron Microscopy Sciences, Hatfield, PA, USA; catalog #14800) or FLUORO-GEL with Tris buffer (Electron Microscopy Sciences, Hatfield, PA, USA; catalog #17983-100). Sample size was not determined using a priori power analysis but was based on the criteria for significance in observations. A total of 8 cerebella from control, 5 from *Car8^wdl/wdl^*, and 5 from *β1-AR^−/−^;β2-AR^−/−^* mice were used in this study, which were processed for immunohistochemistry to examine the expression of β_1_ and β_2_ adrenergic receptor expression throughout the cerebellar cortex.

### 2.11. Imaging of Immunostained Tissue Sections

Photomicrographs of stained tissue sections were captured with Zeiss AxioCam MRm (fluorescence) and AxioCam MRc5 (DAB-reacted tissue sections) cameras mounted on a Zeiss Axio Imager.M2 microscope or on a Zeiss AXIO Zoom.V16 microscope. Apotome imaging (Apotome.2, Zeiss) of tissue sections was performed and images acquired and analyzed using either Zeiss AxioVision software (release 4.8) or Zeiss ZEN software (Zeiss, Jena, Germany; 2012 edition). After imaging, the raw data was imported into Photoshop CC 2022 (Adobe, Mountain View, CA, USA; 2022 version) and corrected for brightness and contrast levels.

### 2.12. Data Analyses

We used a repeated measures two-way ANOVA (*p* < 0.05) followed by a Tukey’s multiple comparisons test to compare data between control and *Car8^wdl/wdl^* mice, before and after propranolol treatment. We used a paired *t*-test (*p* < 0.05) to compare *β1-AR^−/−^;β2-AR^−/−^* and *β1-AR^−/−^;β2-AR^−/−^;Car8^wdl/wdl^* mice before and after propranolol treatment. We used PRISM for two-way ANOVA and *t*-test analyses. We fitted a linear regression model between the mean cerebellar nuclei neuron firing parameters (firing rate, mode ISI^−^^1^, CV, and CV2) and mean average tremor power in each group. We used the MATLAB function ‘fitlm’ to calculate the fit and statistical significance of the fit for each parameter separately. The number of animals tested is represented by ‘N’; the number of cells included in the analyses of electrophysiological recordings is represented by ‘n’. ‘Control’ refers to C57BLKS/J mice. *p* value > 0.05 = ns, ≤0.05 = *, ≤0.01 = **, ≤0.001 = ***, <0.0001 = ****.

### 2.13. Data Visualization

The raw tremor traces and electrophysiological recordings included in the figures are 5 s-long traces from a representative mouse or cellular recording. The insets of representative electrophysiological recordings are 500 ms long. We calculated the mean firing rate at each spike in the visualized spike trains by calculating the firing rate (spikes/s) in the previous 0.5 s. The height of the bar-graphs represents the mean for each group and the individual replicates are visualized using dots, squares, or triangles. Most figure colorizations in this article are from the Wong, 2011 [83] palette optimized for colorblind individuals. The schematics were drawn in Illustrator CC 2022 (Adobe, Mountain View, CA, USA; 2022 version) and then imported into Photoshop to construct the full image panels.

## 3. Results

### 3.1. Propranolol Reduces Tremor in Control and Car8^wdl/wdl^ Mice

Although the cerebellum is increasingly implicated in tremor pathology, it remains unclear whether propranolol acts through the cerebellar circuit to reduce tremor. We measured tremor power using our custom-built tremor monitor in which mice can move around freely (Figure 1A). Control mice exhibit a physiological tremor with a peak amplitude between 8–10 Hz (Figure 1B,C). *Car8^wdl/wdl^* mice also exhibit a tremor between 8–10 Hz (Figure 1B,C), but with a higher amplitude than in control mice, suggesting that these mutants have a pathophysiological tremor. We found that tremor in control and *Car8^wdl/wdl^* mice was significantly reduced following treatment with a clinically relevant 20 mg/kg dosage of propranolol (Figure 1B–D). After propranolol treatment, tremor in *Car8^wdl/wdl^* mice was similar to what we observed in control mice at baseline. The means, SEM, and specific *p*-values for data in all figures are listed in Appendix A.

Because we observed the peak tremor frequency at the frequency of physiological action tremor, we next set out to investigate whether the reduction in tremor severity was caused by a decrease in movements. We used our tremor monitor recordings to detect non-tremor movements (Figure 1E). We found that the number of non-tremor movements during the tremor monitor recordings was not significantly different between control and *Car8^wdl/wdl^* mice, before and after propranolol treatment (Figure 1F). These data indicate that propranolol is specifically effective at reducing physiological tremor in control mice and pathophysiological tremor in *Car8^wdl/wdl^* mice.

### 3.2. Propranolol Does Not Impact General Locomotor Behaviors

Although propranolol is effective in reducing tremor, its potential effects on other motor behaviors in mice are unclear. One potential consideration is that propranolol works specifically on tremor and is ineffective on the other types of movement disorders that also involve cerebellar circuit function. Notably, in addition to tremor, *Car8^wdl/wdl^* mice also exhibit ataxia and dystonia [63]—motor disorders that have both been shown to involve cerebellar defects. To determine whether the previously demonstrated therapeutic effects of propranolol are possibly specific to tremor in *Car8^wdl/wdl^* mice, we conducted the open field and footprinting assays. The open field assay (Figure 2A) gives an indication of overall locomotion patterns. Data from control and *Car8^wdl/wdl^* mice without propranolol is newly analyzed from the dataset published in Miterko et al., 2021, whereas data from control and *Car8^wdl/wdl^* mice that have been treated with propranolol was obtained during this study. Consistent with our findings that propranolol does not affect the non-tremor movement index during tremor measurements (Figure 1E,F), we found that movement time, movement episodes, and ambulatory activity were unaffected by propranolol treatment in control and *Car8^wdl/wdl^* mice (Figure 2B). This confirms that although propranolol reduces tremor in both conditions (Figure 1B–D), it does not affect overall locomotion.

We next used footprinting analysis for the examination of forepaw gait parameters during locomotion; stride, sway, and stance. Stride lengths (the distance between steps of the same paw) reflect the overall mobility of a subject, with reduced lengths representing reduced mobility. Stance distance (the hypotenuse of stride and sway) represents posture during gait. Sway distances (the distance between left and right paw placement) indicate the stability of a subject’s balance in gait. We found that *Car8^wdl/wdl^* mutants have no significant stride or stance length differences compared to control mice (Figure 2D, left and center), indicating a similar overall degree of mobility. However, *Car8^wdl/wdl^* mice exhibit a significantly increased forepaw sway distance compared to control mice, which reflects their characteristic imbalanced, “waddling” gait (Figure 2D, right). Higher or irregular sway distance represents ataxic gait patterns and is strongly correlated with high fall risk in human patients as well [84]. Notably, this pathological gait pattern does not change in *Car8^wdl/wdl^* mice after propranolol treatment, which is reflected in the lack of change in sway distance. These results suggest that propranolol’s therapeutic effects are likely specific to tremor and may not affect other aspects of the motor pathophysiology observed in *Car8^wdl/wdl^* mice.

### 3.3. Propranolol Modulates Purkinje Cell and Cerebellar Nuclei Neuron Firing Activity

The neural mechanisms of propranolol are unclear. Since propranolol specifically affects tremor but not ataxic gaits in *Car8^wdl/wdl^* mice, this may suggest that any changes in neural circuit activity that propranolol induces are also likely specific to hallmarks of tremor pathophysiology. We sought to find potential cerebellar circuit mechanisms of propranolol by conducting awake, single-unit in vivo electrophysiological recordings of both Purkinje cells and cerebellar nuclei neurons in control and *Car8^wdl/wdl^* mice before and after propranolol treatment (Figure 3A and Figure 4A). Purkinje cell misfiring has been shown to be a key driver of tremor pathophysiology, and the induction of rhythmic bursting activity patterns in Purkinje cells using the drug harmaline or optogenetic stimulation causes strong tremor phenotypes in mice [60]. Purkinje cells from healthy control mice fire consistently, whereas *Car8^wdl/wdl^* Purkinje cells have a high-frequency bursting pattern (Figure 3B). Purkinje cells fire two different types of action potentials—simple spikes (spontaneously occurring) and complex spikes (driven by climbing fiber input from the inferior olive). Delineating the activity patterns of these action potentials provides a key readout of cerebellar circuit function (Figure 3A). We reasoned that examining Purkinje cell activity changes induced by propranolol would reveal circuit-level aspects of how the drug affects motor function to reduce tremor. We found that following propranolol treatment, the simple spike firing rates of both control and *Car8^wdl/wdl^* Purkinje cells were significantly decreased (Figure 3B,C). Despite the high-frequency bursting pattern of *Car8^wdl/wdl^* Purkinje cells, the gaps in activity between bursts bring down the overall firing rate when averaged over the recording duration. We noted that the “within burst” activity of *Car8^wdl/wdl^* Purkinje cells after propranolol appears remarkably similar to that of control Purkinje cells at baseline (Figure 3B, compare blue to pink trace). We quantified the mode of the inverse interspike intervals (ISI^−1^) of each cell to better compare the patterns of control and *Car8^wdl/wdl^* Purkinje cells during active firing periods and minimize the skewing effects of inter-burst pauses in *Car8^wdl/wdl^* cells. Interestingly, the mode ISI^−1^ of *Car8^wdl/wdl^* simple spikes after propranolol administration was significantly reduced to the same level of the control animals at baseline, mirroring the tremor findings in Figure 1 (Figure 3C). Next, we quantified both the coefficient of variance (CV) and CV2 to examine firing irregularities across conditions. CV is a measure of irregularity of interspike intervals over the entire recording; a higher CV value indicates a greater overall bursting firing pattern in the cell’s activity. It is therefore expected that simple spike CV in *Car8^wdl/wdl^* Purkinje cells is significantly higher than in controls; however, CV is not significantly changed by propranolol in both control and *Car8^wdl/wdl^* (Figure 3C). CV2 is a measure of irregularity of directly adjacent interspike intervals [78] and can be used to detect erratic neural activity in cerebellar mutants [72,75]. We found that CV2 is also not significantly changed by propranolol in either control or *Car8^wdl/wdl^* mice. Altogether, this data shows that propranolol affects firing rates, but not firing irregularity, in Purkinje cell simple spikes.

For Purkinje cell complex spikes, both the firing rate and mode ISI^−1^ were significantly decreased in control and *Car8^wdl/wdl^* mice after propranolol treatment (Figure 3D). The complex spike CV was significantly decreased after propranolol in *Car8^wdl/wdl^* mice only. The reduced *Car8^wdl/wdl^* complex spike CV following propranolol is not significantly different from the control complex spike CV at baseline. CV2 measures are not changed by propranolol in either genotype. These quantifications indicate that propranolol reduces the firing rate and corrects some features of irregularity of Purkinje cell complex spikes.

We next examined the electrophysiological effects of propranolol on cerebellar nuclei neurons (Figure 4A), which are direct downstream targets of Purkinje cells and form the predominant output from the cerebellum to other regions in the motor circuit. We recorded from cells in the interposed nuclei (labeled as IN in the figures), which control ongoing motion through connections with regions such as the red nucleus and thalamus. Recent studies show that DBS targeted to the interposed nucleus alleviates tremor in mouse models [60,72], and patterned stimulation of this region can cause strong tremor in mice [60]. We reasoned that studying the electrophysiological effects of propranolol in the interposed nucleus could, therefore, provide important insights into the potential neural mechanisms of propranolol at key nodes of the motor circuit. We found that propranolol significantly reduces the firing rate of cerebellar nuclei neurons in both control and *Car8^wdl/wdl^* mice (Figure 4C). *Car8^wdl/wdl^* cerebellar nuclei neurons exhibit a high-frequency bursting pattern similar to their Purkinje cells (Figure 4B), which is reflected in their high mode ISI^−1^ and CV compared to controls. Propranolol corrects both the high mode ISI^−1^ and CV, which are significantly reduced in *Car8^wdl/wdl^* cerebellar nuclei neurons to control levels after treatment (Figure 4C). CV2 measures are unchanged in cerebellar nuclei neurons in both control and *Car8^wdl/wdl^* mice following propranolol treatment.

Finally, we investigated which of the parameters that describe the firing patterns in cerebellar nuclei neurons could best predict tremor severity. We fitted a linear regression model between the group means in firing rate, mode ISI^−1^, CV, and CV2, and group mean in maximal tremor power. Only the model fit between mode ISI^−1^ and maximal tremor power was significant. This analysis indicates that propranolol may reduce the power of tremor by reducing the predominant firing rate of cerebellar nuclei neurons.

### 3.4. β1 and β_2_ Adrenergic Receptors Are Expressed throughout the Cerebellar Cortex

Previous studies have indicated that β_1_ and β_2_ adrenergic receptors, the primary molecular targets of propranolol, are expressed throughout many regions of human and rodent brains [85] and are functionally active at several synapses within the cerebellar circuit [86,87]. Radiographic studies have visualized β_1_ and β_2_ receptors in the cerebellum on a broad scale [88,89]; however, the anatomical distribution and organization of these receptors throughout the cerebellar circuit remain unclear. Here, we used immunohistochemistry with commercial β_1_ and β_2_ adrenergic receptor antibodies to understand the possible localization of propranolol’s targets throughout the cerebellar cortex. We found antibody binding throughout the cerebellar cortex (Figure 5A,E), present in the granular layer, Purkinje cell layer, and molecular layers. Co-labeling with calbindin revealed that the β_1_ and β_2_ antibody signal is present in Purkinje cell bodies and dendrites (Figure 5B–H″). We next sought to determine whether *Car8^wdl/wdl^* mice exhibit differences in β_1_ and β_2_ antibody binding compared to control mice, given their known cerebellar pathology. Comparison of control and *Car8^wdl/wdl^* tissue show no disparity in β_1_ and β_2_ signal, with expression also seen in the granular layer, Purkinje cell layer, and molecular layers for *Car8^wdl/wdl^* mice as in the controls (Appendix A). The localization of β_1_ and β_2_ adrenergic receptor antibody binding in the cerebellum supports our findings that propranolol may modulate neural activity at key nodes of the cerebellar circuit during tremor reduction in this model.

### 3.5. Propranolol Acts through β_1_ and β_2_ Adrenergic Receptors to Reduce Tremor

We sought to determine whether our tremor and electrophysiology findings were dependent on β_1_ and β_2_ receptors during propranolol treatment. To study this, we used *β1-AR^−/−^;β2-AR^−/−^* double homozygous mutant mice with non-functional β_1_ and β_2_ receptors [67]. The genetic strategy used to generate these mice resulted in deletions of specific transmembrane segments of the *β_1_* and *β_2_* receptor complexes, suggesting that the receptors are rendered as non-functional [67,90,91]. The commercial, polyclonal *β_1_* and *β_2_* receptor antibodies used here may detect the truncated, mutant *β_1_* and *β_2_* receptor in *β1-AR^−/−^;β2-AR^−/−^*, which could explain the persistent signal we observed when comparing the mutants to control and *Car8^wdl/wdl^* mice (Appendix A). For our study, we therefore consider the mutants as loss of function mutants.

Using *β1-AR^−/−^;β2-AR^−/−^* mice, we conducted tremor, footprinting, and electrophysiological measurements. We reasoned that if propranolol acts through β_1_ and β_2_ receptors to elicit the reduction in tremor, then *β1-AR^−/−^;β2-AR^−/−^* mice would have no change in tremor amplitude measurements following propranolol treatment. Indeed, tremor monitor recordings show that *β1-AR^−/−^;β2-AR^−/−^* mice have no significant difference in tremor before or after propranolol (Figure 6A–C), which is in contrast to the effect of propranolol on tremor in control and *Car8^wdl/wdl^* mice (Figure 1). Non-tremor movement analysis shows that there is also no significant difference in movements following propranolol in *β1-AR^−/−^;β2-AR^−/−^* mice (Figure 6D). Furthermore, propranolol does not impact the stride, stance, or sway characteristics of *β1-AR^−/−^;β2-AR^−/−^* mice in the footprinting assay, mirroring the finding that propranolol does not alter gait parameters in control or *Car8^wdl/wdl^* mice (Figure 2).

Curiously, our electrophysiological recordings in *β1-AR^−/−^;β2-AR^−/−^* mice show that propranolol is able to alter some aspects of Purkinje cell activity despite the deletion of functional domains in *β_1_* and *β_2_* receptor in the mutant mice (Figure 6G–I). In *β1-AR^−/−^;β2-AR^−/−^* Purkinje cell simple spikes, the firing rate is significantly decreased after propranolol, although mode ISI^−1^, CV, and CV2 measures are unchanged after propranolol (Figure 6H). Complex spike activity is also altered by propranolol in *β1-AR^−/−^;β2-AR^−/−^* mice, with firing rate and mode ISI^−1^ significantly decreased while CV and CV2 are significantly increased (Figure 6I). Nevertheless, the changes in Purkinje cell spiking activity are not propagated to cerebellar nuclei neurons, as we do not observe a change in cerebellar nuclei neuron firing rate, mode ISI^−1^, CV, or CV2 after propranolol treatment (Figure 6J,K). These data further support the hypothesis that changes in cerebellar nuclei neuron activity are necessary for changes in tremor severity.

To further understand the involvement of β_1_ and β_2_ receptor function in propranolol-mediated tremor reduction, we crossed our *Car8^wdl/wdl^* and *β1-AR^−/−^;β2-AR^−/−^* lines to generate *β1-AR^−/−^;β2-AR^−/−^;Car8^wdl/wdl^* triple mutant mice. These mice lack CAR8 protein in their Purkinje cells but additionally lack β_1_ and β_2_ receptor function. We reasoned that examining the effects of propranolol in these *β1-AR^−/−^;β2-AR^−/−^;Car8^wdl/wdl^* triple mutant mice would provide further evidence of propranolol acting through β_1_ and β_2_ receptors to reduce tremor, since these mice persist with the pathophysiological *Car8^wdl/wdl^* tremor. We found that in *β1-AR^−/−^;β2-AR^−/−^;Car8^wdl/wdl^* mice, propranolol has no effect on tremor or non-tremor movements (Figure 7A–C). Additionally, electrophysiology of the cerebellar nuclei in *β1-AR^−/−^;β2-AR^−/−^;Car8^wdl/wdl^* mice shows no change in firing rate, mode ISI^−1^, CV, and CV2 following propranolol treatment (Figure 7D,E). Altogether, the data from *β1-AR^−/−^;β2-AR^−/−^* double mutants and *β1-AR^−/−^;β2-AR^−/−^;Car8^wdl/wdl^* triple mutants indicate that propranolol modulates cerebellar function, and tremor severity, through β_1_ and β_2_ receptors.

## 4. Discussion

The continued use of propranolol as a treatment for tremor necessitates investigations into its in vivo neural mechanisms. In this study, we found that propranolol reduces tremor in *Car8^wdl/wdl^* mice that have pathophysiological tremor that arises as a result of Purkinje cell dysfunction. We then uncovered that propranolol changes cerebellar circuit activity patterns during tremor reduction in *Car8^wdl/wdl^* mice, mainly by slowing firing rates and correcting firing irregularity in Purkinje cells and cerebellar nuclei neurons. Our findings provide evidence that propranolol’s potential tremor reduction mechanism of action is through influencing central nervous system activity. We show that the cerebellum, which has critical roles in tremor pathophysiology, is impacted by propranolol treatment on a circuit level.

In addition to treating pathophysiological tremor in *Car8^wdl/wdl^* mice, we also found that propranolol reduces physiological tremor in control mice. This result corresponds with the known effects and usages of propranolol in humans [92]. Propranolol’s ability to reduce both physiological and pathophysiological tremor suggest that its targets reside in a tremor-causing circuit. Our findings that other locomotor behaviors, as measured by open field, footprinting, and non-tremor movement analyses, were unaffected by propranolol also corroborates this idea. Tremor recordings and subsequent analyses of these oscillating signals can include harmonics. Harmonics occur at frequencies that are multiples of the base frequencies. For example, an oscillation at 10 Hz may produce harmonics at frequencies such as 20 and 30 Hz. Specific characteristics within tremor motion detection, such as the presence of harmonics, can inform on the differential identification of tremor subtypes [93,94,95]. The identification of harmonics is relatively straightforward if the base oscillation is at a single frequency. However, the composition of the tremor power spectrum in our cohort shows multiple elevated frequencies (Figure 1C, Figure 6B, Figure 7A and Appendix A). Although the predominant elevated frequency is ~10 Hz, we also see elevation of the curve at ~13 Hz and ~17 Hz for all groups tested. These frequencies may represent the contributions of different muscles and/or different types of movements to the overall tremor spectrum [96,97]. Additionally, both *Car8^wdl/wdl^* and physiological tremors are action tremors, which means that there is an increase in tremor severity with movement [63,98]. The result is that, at the moment at which we intend to detect movement, by necessity tremor must also occur. This has the potential to produce harmonics in the frequency range that we use to estimate non-tremor movement events. Therefore, we cannot exclude the possibility of tremor harmonics occurring within the 50 to 100 Hz range used for the “non-tremor” analysis. However, we do not observe strong responses at these higher frequencies, and any contribution to non-tremor movements would likely be minimal (see Appendix A). Moreover, it is useful to view the non-tremor movement analysis alongside our open field analysis (Figure 2A,B). We find that *Car8^wdl/wdl^* and control mice have unchanged open field assay measurements after propranolol administration and are similar to each other at baseline, the only difference being the ambulatory activity count, which is reduced in *Car8^wdl/wdl^* mice at baseline, relative to control mice. If a change in movement alone accounted for the increased tremor we find in *Car8^wdl/wdl^* mice relative to control mice at baseline, we would expect that *Car8^wdl/wdl^* mice would move more than control mice at baseline. Similarly, if a change in movement alone accounted for the decreased tremor severity we find in *Car8^wdl/wdl^* mice after propranolol administration relative to baseline, we would expect that *Car8^wdl/wdl^* mice with propranolol would move less than *Car8^wdl/wdl^* mice at baseline. However, with the open field assay we see the opposite between the genotypes at baseline and no change within the genotypes between baseline and with propranolol, suggesting that changes in movement are not driving the differences in tremor severity. This is supported with our non-tremor movement analysis. Frequencies from 50 to 100 Hz include sharp deflections of acceleration indicative of initiation of movement or footfalls in addition to possible harmonics of underlying tremor. Therefore, we use this range as an estimate of movement within the tremor monitor. Similar to our open field findings, we see no difference between *Car8^wdl/wdl^* and control mice at baseline. Additionally, we see no difference between *Car8^wdl/wdl^* mice at baseline and *Car8^wdl/wdl^* mice after propranolol administration. Taken together, these data suggest that our tremor findings are not driven by changes in overall movement. Furthermore, we found that β_1_ and β_2_ adrenergic receptors, thought to be the primary molecular targets of propranolol, are broadly labeled throughout the cerebellar cortex and are required for propranolol to reduce tremor in mice. These data provide evidence that β_1_ and β_2_ receptor expression help define the tremor circuit, adding to our understanding of which brain regions mediate tremor.

Our findings that propranolol significantly reduces tremor in *Car8^wdl/wdl^* mice are in agreement with previous studies showing propranolol’s efficacy in treating human patients with tremor [43,45,99] and mouse models of tremor, including harmaline-induced tremor and the *α1^–/–^* mouse model with depleted GABAA receptor α1 subunits [100,101,102]. Additionally, our evidence that propranolol reduces physiologic tremor in control mice indicates that propranolol even has a robust effect on normal circuit function. Together, our data showing that propranolol works to treat cerebellar-based tremor pathophysiology and reduce physiological tremor support the use of propranolol in mouse models to further understand how tremor and propranolol mechanistically intersect in specific cells.

Previous studies have also examined the effects of propranolol in human patients with ataxia and dystonia, which is interesting given the potential common origins of these diseases from within the cerebellum. Propranolol was found to temporarily alleviate ataxia in a few clinical reports [103,104]. Additionally, differential effects of propranolol were found in patients with comorbid dystonia and tremor, suggesting that at least some aspects of the pathophysiology underlying these two conditions are distinct [105]. Our findings that propranolol does not affect gross locomotor behavior in control, *Car8^wdl/wdl^*, or *β1-AR^−/−^;β2-AR^−/−^* mice in open field and footprinting assays—and especially that propranolol does not affect ataxic gait parameters in *Car8^wdl/wdl^* mice—supports the hypothesis that propranolol and its mechanisms are specific to tremor pathophysiology. However, additional motor assays that are able to differentially assess ataxic phenotypes in *Car8^wdl/wdl^* mice, such as the rotarod or parallel floor rod assays, could be conducted to fully evaluate this possibility. Still, the source of ataxia, dystonia, and tremor may all intersect at a shared anatomical circuit in the cerebellum [106,107,108,109,110,111,112]. In *Car8^wdl/wdl^* mice specifically, ataxia, dystonia, and tremor result from a single mutation affecting cerebellar Purkinje cells [63]. Interestingly though, it could be that there are separate mechanisms for the inception and propagation of faulty neural signals in these disorders, as evidenced by differential activity signatures within this shared circuit [60,66,72,77]. Indeed, our electrophysiology findings showing that propranolol changes specific parameters of Purkinje cell and cerebellar nuclei neuron firing activity while reducing tremor in control and *Car8^wdl/wdl^* mice support this hypothesis. The pathophysiological baseline firing activity of *Car8^wdl/wdl^* mice has been well-documented and is characterized by high-frequency burst periods [63,66,69]. Correspondingly, high-frequency burst activity in both Purkinje and cerebellar nuclei neurons, as induced by harmaline or optogenetic stimulation, has been shown to cause tremor in mice [60]. Our finding that propranolol resolves *Car8^wdl/wdl^* tremor while reducing the firing rate, mode ISI^−1^, and CV irregularity of Purkinje cells and cerebellar nuclei neurons within the high-frequency bursts, is interesting in the context of those studies. Yet, it also adds to the growing body of evidence showing that the targeted therapeutic manipulation of cerebellar circuit activity patterns, such as with DBS, has strong potential for treating motor disorders [60,69]. Furthermore, our data showing that propranolol lowers the firing rate of Purkinje cells and cerebellar nuclei neurons in control mice with reduced physiological tremor provides insight into what one might expect from an effective therapy; a return to “normal” circuit function may not be the only way to improve behavior to enhance quality of life.

Although we provide additional insight for how the cerebellar circuit contributes to tremor, our study focused on testing how propranolol reduces tremor. Despite being a foremost tremor treatment, a debate still continues about how propranolol might affect the central nervous system. Some studies have postulated that propranolol may have a generally depressive effect in the central nervous system, based on clinical observations and self-reported patient measures of energy and mood levels [43,48] as well as behavioral assays following propranolol treatment in rodents [49,51]. One study reported no change in central nervous system activity following propranolol treatment in healthy human subjects as measured by EEG recordings [42]. Our electrophysiological findings provide compelling evidence that propranolol does indeed have a measurable effect in the central nervous system. It is possible that if the reduction in firing rate caused by propranolol in cerebellar cells is an effect that extends to other brain regions involved in mood regulation, one could postulate that it may have a similar depressive effect on a patient’s energy or mood. Additionally, it is possible that the neural circuit changes we found to be elicited by propranolol occur on a broad scale that could be examined using cortical EEG, which could then be used to compare to healthy subjects who do not have tremor. Recent studies have investigated additional molecular mechanisms of propranolol in the brain, finding evidence that propranolol can inhibit protein synthesis and induce nitric oxide and hydrogen peroxide secretion in certain brain regions [113,114,115]. There is also new evidence that propranolol induces transcriptomic changes in several genes tied to essential tremor etiology [116]. These studies demonstrate that propranolol may affect complex signaling cascades involving molecular pathways that influence myriad intracellular processes [117,118], supporting the notion that propranolol likely has complex effects on brain biochemistry beyond immediate effects of beta-adrenergic blockade.

In addition to altering a single component of the cerebellar circuit, it is also possible that propranolol can directly affect several components of the circuit at the source, including the inferior olive and cerebellar nuclei, or other regions of the motor circuit. Climbing fiber signals originating from the inferior olive may be dampened by the effects of propranolol within the olive itself, which in turn may contribute to the reduction in complex spike activity seen in our recorded Purkinje cells. Complex spike activity patterns and synchrony are increasingly understood to contribute to tremor outcomes, a hypothesis that is supported by evidence in which heightened inferior olive coupling and complex spike synchrony may underlie tremor in several drug-based and genetic mouse models [119,120,121]. It is therefore possible that the inhibitory effect we found propranolol to have on complex spikes dampens the impact of synchronous complex spike activity on downstream regions, ultimately leading to reduced tremor amplitudes. Propranolol may also act directly on cerebellar nuclei neurons, with the effects being independent of upstream input from Purkinje cells, which themselves are affected by propranolol.

While considering the sites of propranolol’s action, it is also important to consider how the cerebellar cortex and cerebellar nuclei interact. Indeed, several studies have shown that cerebellar nuclei neuron activity is often both correlated and anti-correlated to Purkinje cell activity, contrary to the parsimonious assumption that the firing rates of cerebellar nuclei neurons are always the inverse of their Purkinje cell inputs [122,123,124]. There is evidence that synchronous Purkinje cell inhibitory inputs to the cerebellar nuclei results in “timed spiking” patterns, wherein there is a predictable increase in cerebellar nuclei neuron spikes immediately following a Purkinje cell stimulus [122,125,126]. It is possible that the inhibitory effects of propranolol on Purkinje cell firing rates decreases this synchrony coding at corticonuclear synapses, leading to weakened communication of tremorgenic signatures within the circuit, an effect possibly compounded by any electrophysiological effects propranolol may directly have on cerebellar nuclei neurons. Whether propranolol may be directly inhibiting individual regions of the motor circuit simultaneously, or whether it may affect only one specific node in the circuit that results in cascading electrophysiological effects, we show compelling evidence that propranolol induces activity changes in the brain that are strong enough to disrupt, or even possibly rescue, the pathological electrophysiological signatures that underlie tremor.

Previous studies that examined the tremor-reducing mechanisms of propranolol focused on measures in the peripheral nervous system, suggesting that propranolol may be active at β-adrenergic receptors in the muscle spindles during tremor attenuation [36,46,47]. These studies could not rule out the central involvement of propranolol; comparatively, we cannot rule out a peripheral involvement of propranolol as a contributing mechanism in our *Car8^wdl/wdl^* tremor treatment. However, the current consensus is that tremor is a disease of the central nervous system, originating within the brain [52,127,128,129,130]. Therefore, any insights into how tremor-reducing drugs such as propranolol may be acting in the central nervous system during tremor attenuation is crucial to human health. It is possible that effective tremor drugs act both at the source (in the brain) and at the ultimately affected regions (in the periphery) to achieve their full therapeutic effects.

Our immunohistochemistry data showing that the β_1_ and β_2_ receptor antibody signal is heavily expressed in the cerebellum circuit provides an anatomical framework for understanding where propranolol may be active. Although numerous studies have examined the distribution and role of β-adrenergic receptors in the rest of the body [131,132,133,134], radiographic studies have reported on the expression of these receptors in the brain, providing a general overview of where the receptors might be located [88,89]. In our study, we visualized β_1_ and β_2_ receptors in the cerebellum to appreciate their cellular resolution, uncovering their distribution across the cerebellar cortical layers and in relation to Purkinje cells. Our findings that these receptors are necessary for propranolol to reduce tremor in *β1-AR^−/−^;β2-AR^−/−^* double-mutant mice and *β1-AR^−/−^;β2-AR^−/−^;Car8^wdl/wdl^* triple mutant mice provide evidence that propranolol acts through β_1_ and β_2_ receptors, and is likely not working through secondary affinities for β_3_ or 5-HT receptors for its therapeutic effects. Our electrophysiology data showing that *β1-AR^−/−^;β2-AR^−/−^* Purkinje cells do change firing activity with propranolol, may however be explained by propranolol’s secondary affinity for 5-HT receptors. These receptors are abundant in the afferent fibers of the cerebellum that directly modulate Purkinje cell activity [135], and we predict that their function is likely unchanged in the *β1-AR^−/−^;β2-AR^−/−^* mice.

Although we did not record from Purkinje cells in *β1-AR^−/−^;β2-AR^−/−^;Car8^wdl/wdl^* triple mutant mice, we surmise that the electrophysiological responses to propranolol would resemble a combination of the Purkinje cell effects seen independently in *Car8^wdl/wdl^* baseline cells, and *β1-AR^−/−^;β2-AR^−/−^* double mutant cells. The characteristic high-frequency bursting signature of *Car8^wdl/wdl^* Purkinje cells is likely to be preserved, but there may be reductions in overall simple and complex spike firing rates such as those seen in *β1-AR^−/−^;β2-AR^−/−^* Purkinje cells following propranolol. Ultimately, our tremor monitor and cerebellar nuclei electrophysiology data in the *β1-AR^−/−^;β2-AR^−/−^* and *β1-AR^−/−^;β2-AR^−/−^;Car8^wdl/wdl^* mice support the hypothesis that the mechanisms underlying propranolol’s tremor reducing effects require intact β_1_ and β_2_ receptors in the cerebellum. Our genetic crosses between the *β1-AR^−/−^;β2-AR^−/−^* and *Car8^wdl/wdl^* mutants generated triple mutant mice that were difficult to breed and maintain especially for in vivo electrophysiological experiments. For these reasons, for the mice that we were able to salvage, we chose to focus our recordings on the cerebellar nuclei given their critical roles in cerebellar output; any potential defects could, in theory, be uncovered by examining this node in the circuit. We found that propranolol does not induce changes in our measurements of firing mode in cerebellar nuclei neurons of triple mutants, suggesting the possibility that β_1_ and β_2_ receptors are functionally necessary for tremor reduction in the *Car8^wdl/wdl^* model. In accordance with these data, based on our cerebellar nuclei neuron electrophysiology data from control and *Car8^wdl/wdl^* subjects, we have confidence that the firing mode remains a reliable predictor of tremor outcomes, and this is reflected in the recorded tremor and electrophysiology data from the triple mutants. Still, additional electrophysiological data from *β1-AR^−/−^;β2-AR^−/−^;Car8^wdl/wdl^* mice would offer additional insights into the intersecting roles of β-adrenergic receptor function within the *Car8^wdl/wdl^* model. Although propranolol seems to require β_1_ and β_2_ receptors for tremor reduction, the mechanism may involve a non-canonical signaling pathway in Purkinje cells.

Our findings from this study add to the growing body of evidence that propranolol has prominent effects in the central nervous system, particularly in the cerebellar circuit. Prior studies have found that propranolol inhibits Purkinje cell GABAergic neurotransmission in slice preparations, suggesting that monoaminergic facilitation of cerebellar activity may play a role in synaptic plasticity and motor coordination [86,136]. Indeed, subsequent studies discovered that blocking β-adrenergic receptor function with propranolol has inhibitory effects on learning and memory acquisition processes in both eyeblink conditioning and forced-swim assays [137,138]. Other studies have found that adrenergic agents inhibit Purkinje cell glutamic acid decarboxylase (GAD67) levels following climbing fiber lesions, and that β-adrenergic receptors mediate excitability increases in hyperpolarization-activated cation channels (HCNs) at the cerebellar basket cell-to-Purkinje cell synapse [87,139]. Thus, understanding the mechanisms of propranolol has wider implications, since it is also used to treat other conditions including high blood pressure, heart conditions, anxiety, and migraines. Our study examined the behavioral and electrophysiological effects of propranolol in a mouse model of tremor based on cerebellar dysfunction, and the data show a tremor-reducing mechanism of propranolol that involves the modulation of cerebellar circuit activity. It will be interesting to see which mechanisms are unique to the cerebellum and which ones are shared with other organ systems in these other conditions.

## Figures and Tables

**Figure 1 cells-11-03889-f001:**
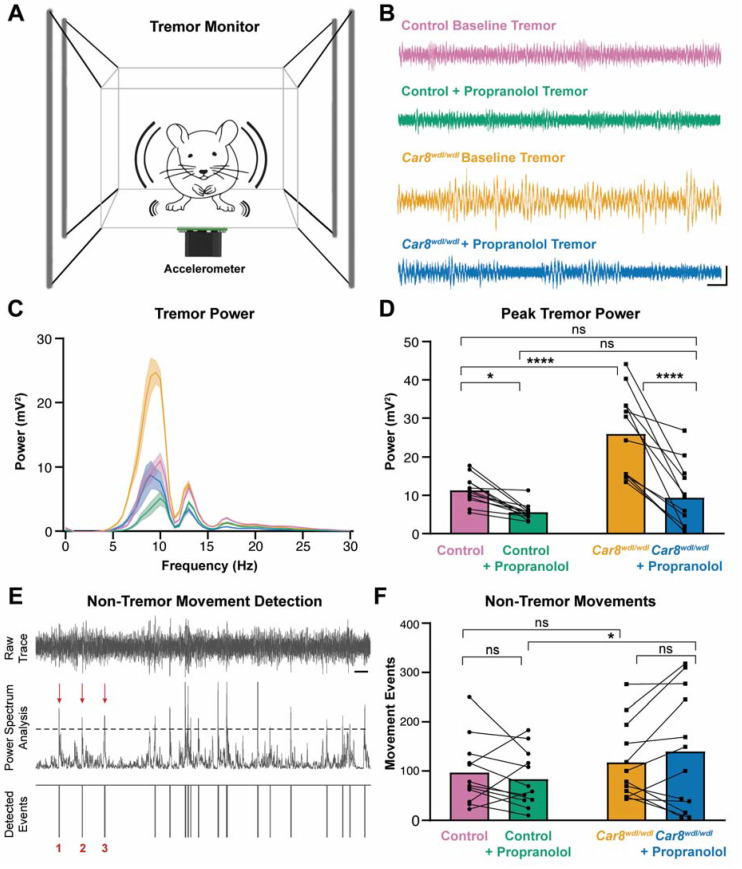
**Propranolol reduces pathological tremor in *Car8^wdl/wdl^* mice and physiological tremor in control mice.** (**A**) Schematic of the tremor monitor configuration. (**B**) Representative raw traces of tremor readings recorded from the tremor monitor for control mice at baseline (pink, N = 12), control mice after propranolol treatment (green, N = 12), *Car8^wdl/wdl^* mice at baseline (orange, N = 12), and *Car8^wdl/wdl^* mice after propranolol treatment (blue, N = 12). Larger vertical deflections indicate stronger tremor power. Scale bar is 50 mV vertical and 500 ms horizontal. (**C**) Line graph depicting tremor power versus frequency. Color representation for groups is maintained from panel (**B**). Power indicates the presence and severity level of tremor, with higher power illustrating stronger severity. Frequency indicates the speed of tremor movements. Following propranolol treatment, both control and *Car8^wdl/wdl^* mice exhibit reduced tremor power compared to baseline. (**D**) Bar graph showing quantifications of peak tremor power at baseline and after propranolol treatment in control and *Car8^wdl/wdl^* mice. Circle and square points represent each individual subject’s peak tremor power for control and *Car8^wdl/wdl^* mice, respectively, with lines connecting each animal’s data before and after treatment. *Car8^wdl/wdl^* mice exhibit significantly stronger peak tremor power at baseline compared to control mice. Following treatment with propranolol, both groups show significantly decreased maximum tremor power. * = *p* < 0.05; **** = *p* < 0.0001; ns = not significant, *p* > 0.05. (**E**) Schematic illustrating the procedure for a non-tremor movement detection analysis within tremor monitor recordings. Scale bar is 1 s. (**F**) Quantifications of non-tremor movements show no significant difference between control and *Car8^wdl/wdl^* mice at baseline, or within groups following propranolol treatment, indicating that tremor severity is not associated with the overall amount of non-tremor movements as captured in the recordings.

**Figure 2 cells-11-03889-f002:**
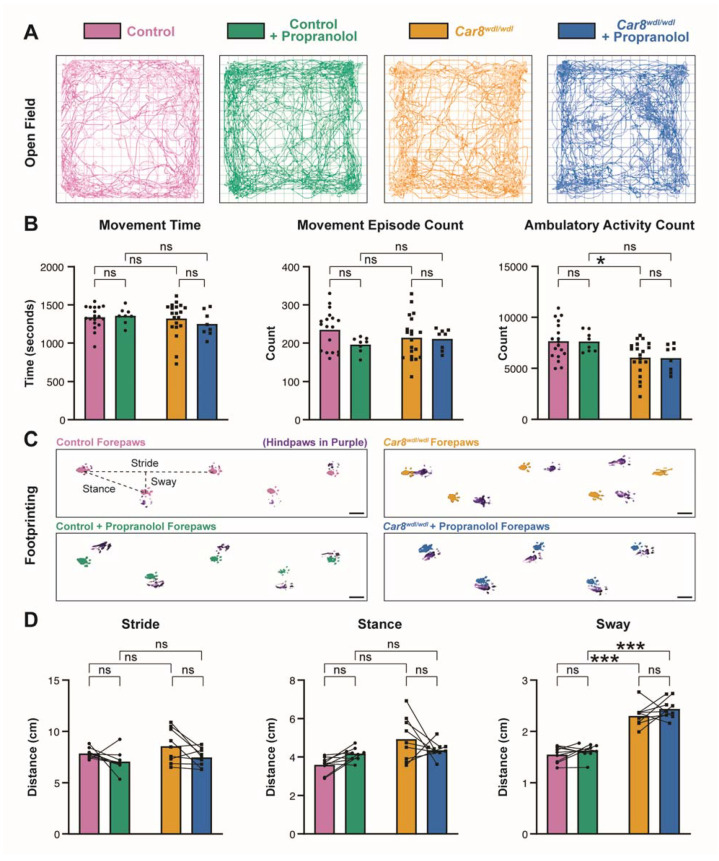
**General activity levels, gross locomotor activity, and gait parameters are unaffected by propranolol.** (**A**) Representative traces of open field activity patterns over a 30 min recording in control mice at baseline (pink, N = 18) and with propranolol (green, N = 8), and in *Car8^wdl/wdl^* mice at baseline (orange, N = 20) and with propranolol (blue, N = 8). Lines indicate the animal’s locomotor trajectory over time. The same color assignment for each group is maintained throughout the remaining figure panels (legend above open field traces). Open field data from control and *Car8^wdl/wdl^* mice without propranolol (pink and orange) is newly analyzed from the dataset published in Miterko et al., 2021. (**B**) Quantifications of open field activity in control and *Car8^wdl/wdl^* mice. There is no significant difference in total movement time or number of movement episodes between control and *Car8^wdl/wdl^* mice at baseline or with propranolol treatment, both within and across groups. No significant difference was found in ambulatory activity with or without propranolol within groups in control or *Car8^wdl/wdl^* mice, or with propranolol between groups. * = *p* < 0.05; *** = *p* < 0.001; ns = not significant, *p* > 0.05. (**C**) Representative traces of forepaw footprinting assays recorded from control (N = 9) and *Car8^wdl/wdl^* mice (N = 9), before and after propranolol. Hindpaw prints are shown in purple for context. The 3 gait parameters assessed are stride (the distance between steps of the same paw), sway (the distance between left and right paw placement), and stance (the hypotenuse of stride and sway). Scale bar is 1 cm. (**D**) Quantifications of footprinting assays for the forepaws. Lines connect before and after treatment data for each animal. There is no significant difference within or between groups in control and *Car8^wdl/wdl^* mice, before and after propranolol, for stride or stance. However, sway distances are significantly increased in *Car8^wdl/wdl^* mice compared to control mice between groups, before and after propranolol. There is no significant sway difference within groups after propranolol treatment compared to baseline levels in both control and *Car8^wdl/wdl^* mice.

**Figure 3 cells-11-03889-f003:**
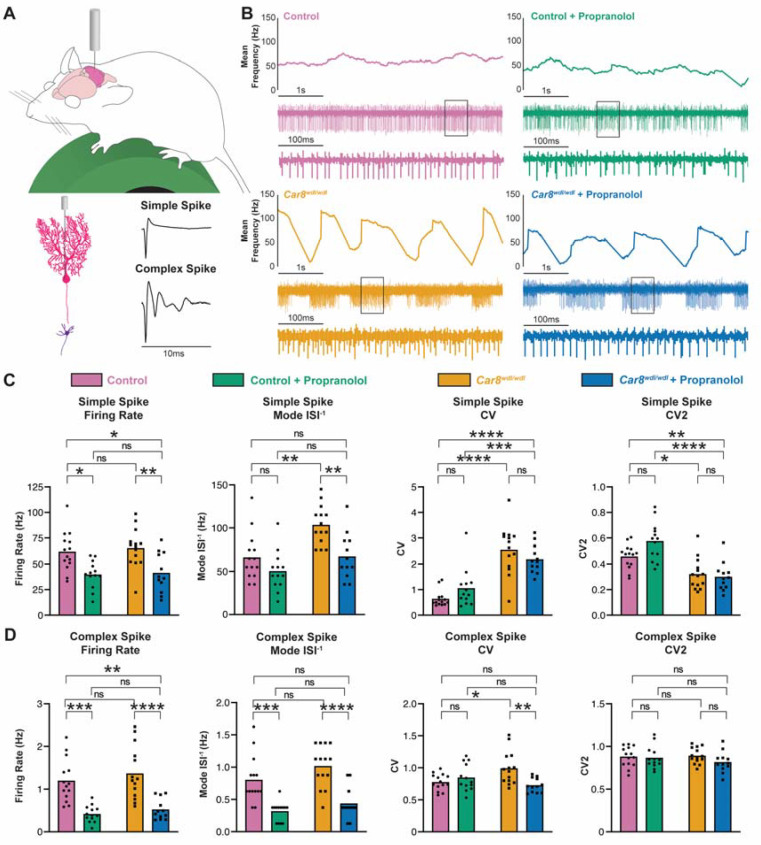
**Propranolol modulates cerebellar Purkinje cell firing activity.** (**A**) Top, schematic of awake in vivo electrophysiology recording setup. Bottom, illustrations of a Purkinje cell (bright pink), and downstream cerebellar nuclei neuron (purple), and the two different types of action potentials Purkinje cells produce—simple spikes and complex spikes. (**B**) Representative raw electrophysiological traces of Purkinje cell activity in control mice before (pink, N = 8, n = 14) and after propranolol treatment (green, N = 5, n = 13), and *Car8^wdl/wdl^* mice before (gold, N = 6, n = 14) and after propranolol treatment (blue, N = 4, n = 12). The line graph shown at the top for each condition represents the mean firing rate in Hz at each point in time for the 5 s spike traces shown below each line graph. Below the 5 s spike traces are magnified views of the spikes within the outlined boxes, spanning 500 ms. The same color assignment for each group is maintained throughout the remaining figure panels (legend above electrophysiology graphs). (**C**) Quantifications of Purkinje cell simple spike firing activity. Propranolol significantly reduces simple spike firing rate in both control and *Car8^wdl/wdl^* Purkinje cells. The mode ISI^−1^ is significantly decreased in *Car8^wdl/wdl^* simple spikes following propranolol treatment but not in controls. Simple spike CV and CV2 measures do not significantly change with propranolol in both control and *Car8^wdl/wdl^* mice. * = *p* < 0.05; ** = *p* < 0.01; *** = *p* < 0.001; **** = *p* < 0.0001; ns = not significant, *p* > 0.05. (**D**) Quantifications of Purkinje cell complex spike firing activity. Propranolol significantly reduces complex spike firing rate in both control and *Car8^wdl/wdl^* Purkinje cells. The mode ISI^−1^ is significantly decreased in both control and *Car8^wdl/wdl^* complex spikes following propranolol treatment. Complex spike CV is significantly reduced in *Car8^wdl/wdl^* mice only, but not in controls. Complex spike CV2 measures do not significantly change with propranolol in both control and *Car8^wdl/wdl^* mice.

**Figure 4 cells-11-03889-f004:**
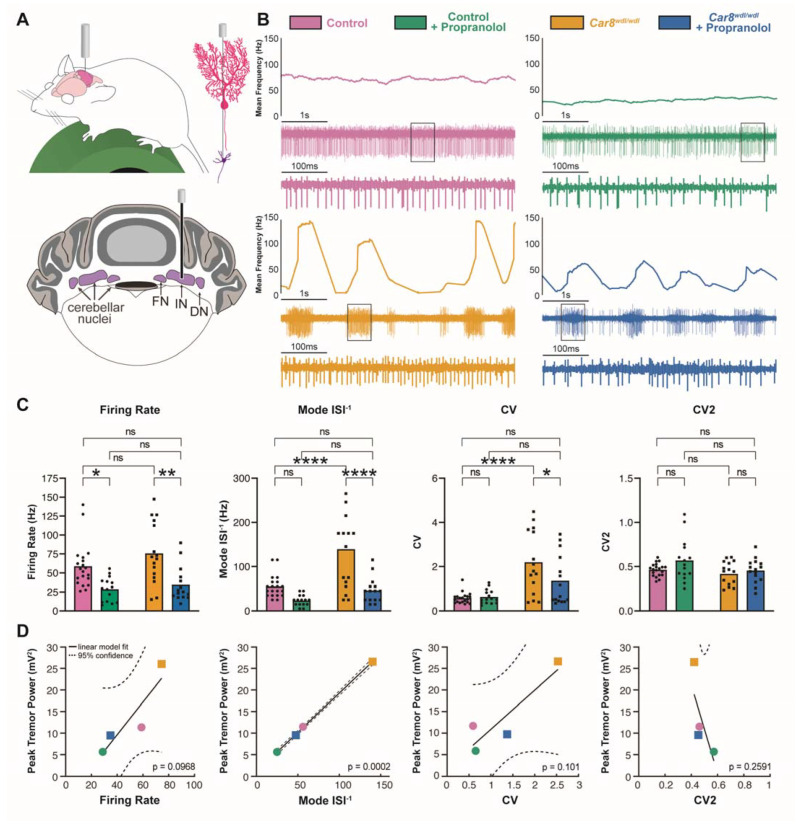
**Propranolol modulates cerebellar nuclei neuron firing activity.** (**A**) Top, schematic of awake in vivo electrophysiology recording setup. Illustrations of a Purkinje cell (bright pink), and downstream cerebellar nuclei neuron (purple) are shown. Bottom, schematic of a coronal mouse cerebellar section with the cerebellar nuclei outlined in purple. Recordings were conducted in the interposed nucleus (IN) as shown with the electrode placement. FN = fastigial nucleus, DN = dentate nucleus. (**B**) Representative raw electrophysiological traces of cerebellar nuclei neuron activity in a control mouse before (pink, N = 6, n = 20) and after propranolol treatment (green N = 6, n = 15), and in a *Car8^wdl/wdl^* mouse before (gold, N = 6, n = 15) and after propranolol treatment (blue, N = 6, n =15). The line graph shown at the top for each condition represents the mean firing rate in Hz at each point in time for the 5 s spike traces shown below each line graph. Below the 5 s spike traces are magnified views of the spikes within the outlined boxes, spanning 500 ms. The same color assignment for each group is maintained throughout the remaining figure panels (legend above electrophysiology graphs). (**C**) Quantifications of cerebellar nuclei neuron firing activity. Propranolol significantly reduces cerebellar nuclei neuron firing rate in both control and *Car8^wdl/wdl^* mice. The cerebellar nuclei neuron mode ISI^−1^ is significantly decreased in *Car8^wdl/wdl^* mice following propranolol treatment but not in controls. Cerebellar nuclei neuron CV is significantly reduced in *Car8^wdl/wdl^* mice only, but not in controls. Cerebellar nuclei neuron CV2 measures do not significantly change with propranolol in both control and *Car8^wdl/wdl^* mice. * = *p* < 0.05; ** = *p* < 0.01; **** = *p* < 0.0001; ns = not significant, *p* > 0.05. (**D**) Linear regression models correlating mean cerebellar nuclei neuron firing parameters (firing rate, mode ISI^−1^, CV, and CV2) and mean average tremor power in each group. Solid black lines indicate linear model fit and dotted black lines indicate 95% confidence intervals. Only the model fit between mode ISI^−1^ and maximal tremor power was significant (*p* = 0.0002).

**Figure 5 cells-11-03889-f005:**
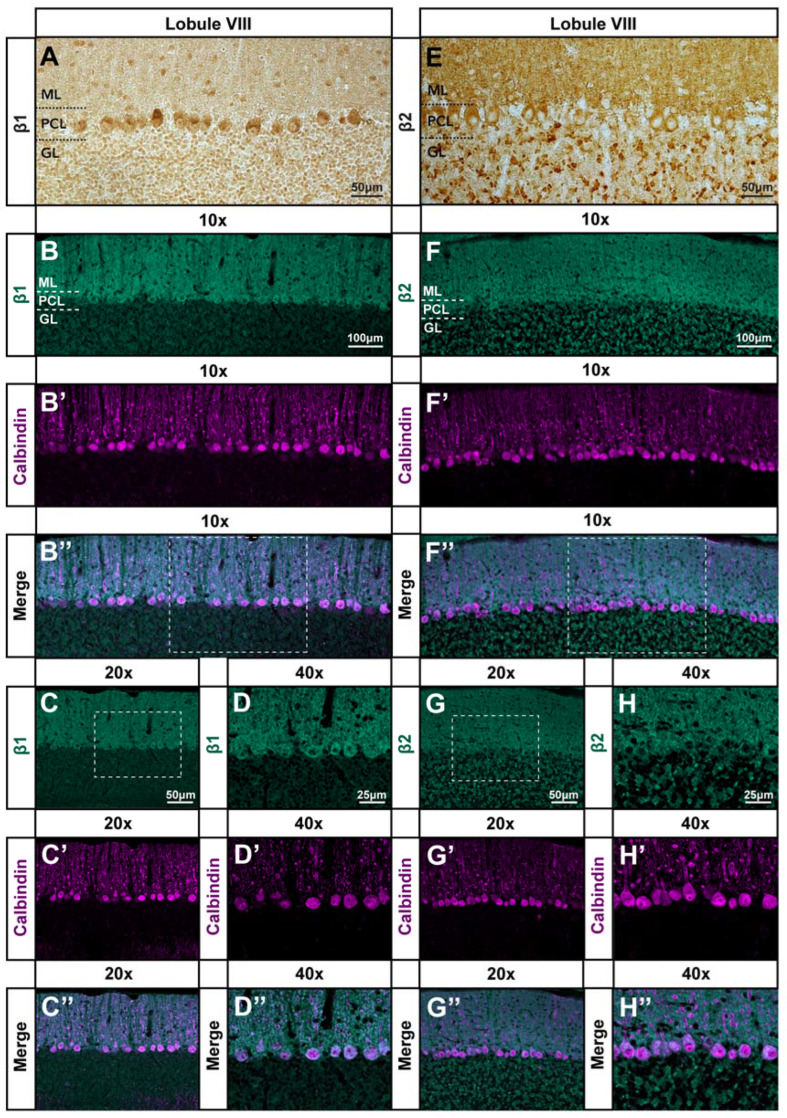
**The β_1_ and β_2_ adrenergic receptor antibody signal is expressed throughout the cerebellar cortex.** (**A**,**E**) Paraffin staining of a coronal section cut through lobule VIII of the cerebellar cortex in control mice (N = 8) showing β_1_ (**A**) and β_2_ (**E**) adrenergic receptor antibody staining in brown. Purkinje cell somata are positioned in the Purkinje cell layer (PCL) underneath the molecular layer (ML), and directly below the PCL lies the granular layer (GL) containing granule cells and various classes of interneurons. The β_1_ and β_2_ signal is expressed throughout all three layers of the cerebellar cortex. Scale bar is 50 μm. (**B**–**B″**,**F**–**F″**) Free-floating fluorescence double staining of the same coronal view of lobule VIII with β_1_ (**B**) and β_2_ (**F**) antibody signal in green, calbindin expression in Purkinje cells in magenta (**B′**,**F′**), and the overlay of β_1_ or β_2_ and calbindin (**B″**,**F″**). Co-localized β_1_ or β_2_ and calbindin expression appears as a brighter, whitish hue. Dotted outlines in B″ and F″ indicate the areas from which the higher magnification images in (**C**–**C″**,**G**–**G″**) were taken. Scale bar is 100 μm. (**C**–**C″**,**G**–**G″**) Higher magnification image of the dotted outlined areas from (**B″**,**F″**) showing β_1_ (**C**) and β_2_ (**G**) signal expression in green, calbindin expression in Purkinje cells in magenta (**C′**,**G′**), and the overlay of β_1_ or β_2_ and calbindin (**C″**,**G″**). Dotted outlines in (**C**,**G**) indicate the area from which the higher magnification images in (**D**–**D″**,**H**–**H″**) were taken. Scale bar is 50 μm. (**D**–**D″, H**–**H″**) Even higher magnification image of the dotted outlined areas from (**C**,**G**) showing β_1_ (**D**) and β_2_ (**H**) signal expression in green, calbindin expression in Purkinje cells in magenta (**D′**,**H′**), and the overlay of β1 (**D″**) or β2 (**H″**) and calbindin. At these higher magnifications, the co-localization of β_1_ or β_2_ and calbindin expression is more easily appreciated. Scale bar is 25 μm.

**Figure 6 cells-11-03889-f006:**
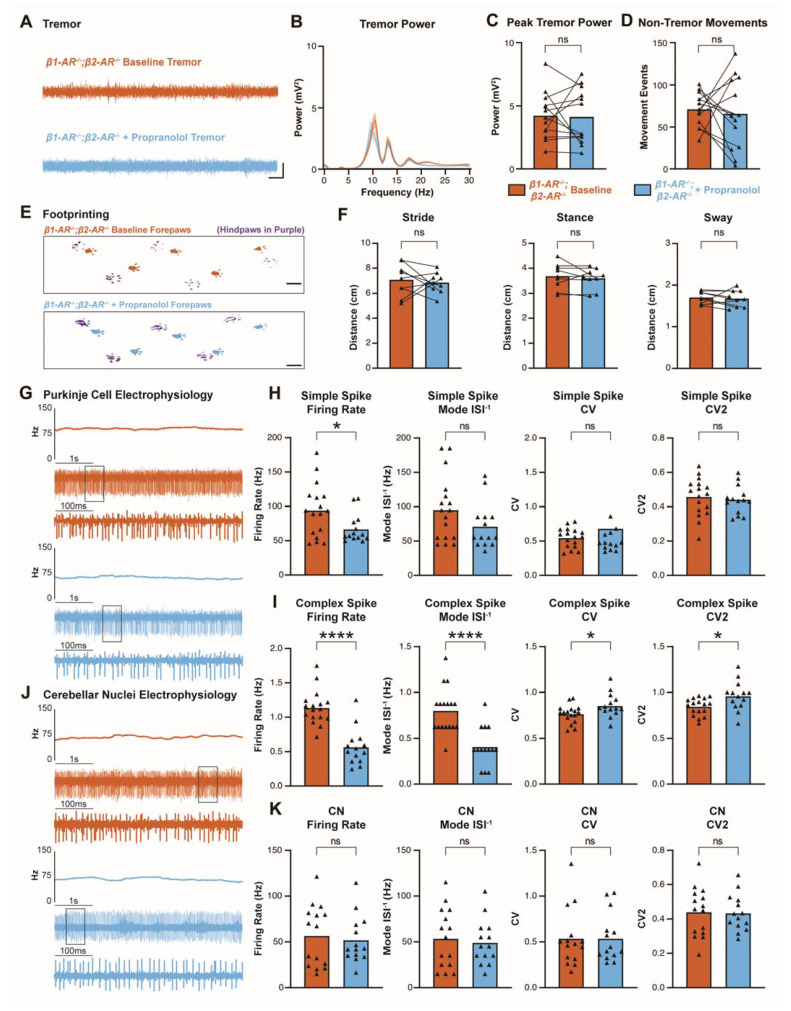
**Propranolol does not modulate tremor, gait, or cerebellar nuclei neuron activity in mice lacking β_1_ and β_2_ adrenergic receptors.** (**A**) Representative raw traces of tremor readings recorded from the tremor monitor for *β1-AR^−/−^;β2-AR^−/−^* mice at baseline (vermillion, N = 11), and after propranolol treatment (sky blue, N = 13). The same color assignment for each group is maintained throughout the remaining figure panels (legend below panels (**C**,**D**)). Scale bar is 50 mV vertical and 500 ms horizontal. (**B**) Line graph depicting tremor power versus frequency. Color representation for groups is maintained from panel **A**. (**C**) Bar graph showing quantifications of peak tremor power at baseline and after propranolol treatment in *β1-AR^−/−^;β2-AR^−/−^* mice with lines connecting each animal’s data before and after treatment. There is no significant difference in peak tremor power after *β1-AR^−/−^;β2-AR^−/−^* mice receive propranolol. * = *p* < 0.05; **** = *p* < 0.0001; ns = not significant, *p* > 0.05. (**D**) Quantifications of non-tremor movements show no significant difference between *β1-AR^−/−^;β2-AR^−/−^* mice at baseline or after being treated with propranolol. (**E**) Representative traces of forepaw footprinting assays recorded from *β1-AR^−/−^;β2-AR^−/−^* mice before and after propranolol (N = 9). Hindpaw prints are in shown in purple for context. Scale bar is 1 cm. (**F**) Quantifications of footprinting assays. Lines connect before and after treatment data for each animal. There is no significant difference before and after propranolol for stride, stance, or sway in *β1-AR^−/−^;β2-AR^−/−^* mice. (**G**) Representative raw electrophysiological traces of Purkinje cell activity in *β1-AR^−/−^;β2-AR^−/−^* mice before (N = 5, n = 17) and after propranolol treatment (N = 4, n = 14). The line graph at the top for each condition represents the mean firing rate in Hz at each point in time for the 5 s spike traces shown below each line graph. Below the 5 s spike traces are magnified views of the spikes within the outlined boxes, spanning 500 ms. (**H**) Quantifications of *β1-AR^−/−^;β2-AR^−/−^* Purkinje cell simple spike firing activity. Propranolol significantly reduces simple spike firing rate in *β1-AR^−/−^;β2-AR^−/−^* mice. Mode ISI^−1^, CV, and CV2 measures are unchanged by propranolol in *β1-AR^−/−^;β2-AR^−/−^* mice. (**I**) Quantifications of Purkinje cell complex spike firing patterns before and after propranolol treatment in *β1-AR^−/−^;β2-AR^−/−^* mice. Propranolol significantly reduces both the firing rate and mode ISI^−1^ of complex spikes in *β1-AR^−/−^;β2-AR^−/−^* mice. Inversely, complex spike CV and CV2 measures are increased following propranolol treatment in *β1-AR^−/−^;β2-AR^−/−^* mice. (**J**) Representative raw electrophysiological traces of cerebellar nuclei neuron activity in *β1-AR^−/−^;β2-AR^−/−^* mice before (N = 6, n = 15) and after propranolol treatment (N = 4, n = 14). (**K**) Quantifications of cerebellar nuclei neuron firing activity. Propranolol has no effect on the firing rate, mode ISI^−1^, CV, or CV2 measures in *β1-AR^−/−^;β2-AR^−/−^* cerebellar nuclei neurons.

**Figure 7 cells-11-03889-f007:**
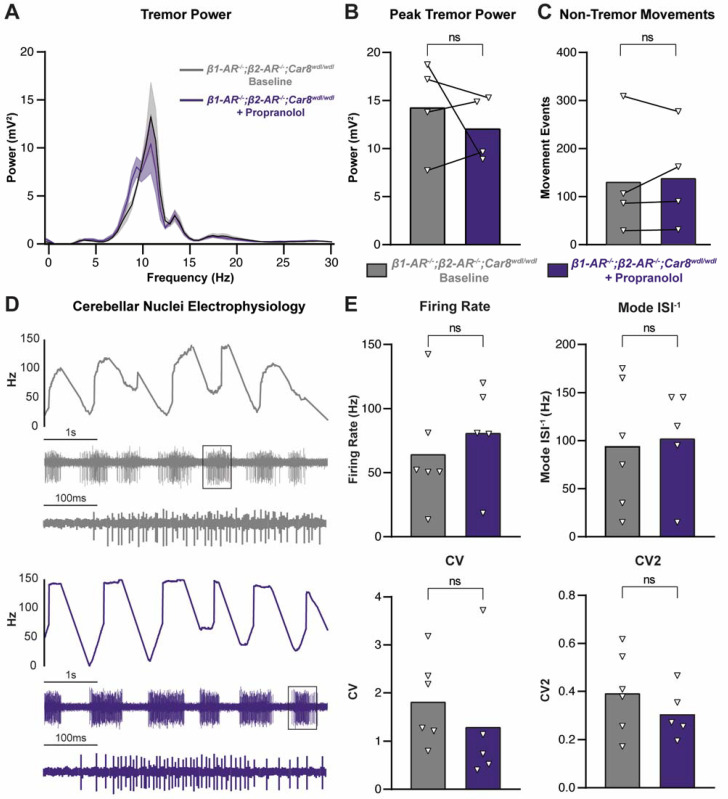
**β_1_ and β_2_ adrenergic receptor function is required for propranolol to reduce *Car8^wdl/wdl^* tremor.** (**A**) Line graph depicting tremor power versus frequency in *β1-AR^−/−^;β2-AR^−/−^;Car8^wdl/wdl^* mice before (gray, N = 4) and after (indigo, N = 4) propranolol treatment. The same color assignment for each group is maintained throughout the remaining figure panels (legend below panels (**B**,**C**)). (**B**) Quantification of peak tremor power in *β1-AR^−/−^;β2-AR^−/−^;Car8^wdl/wdl^* mice, with lines connecting each subject’s before and after propranolol data points. Propranolol does not significantly reduce tremor in *β1-AR^−/−^;β2-AR^−/−^;Car8^wdl/wdl^* mice. ns = not significant, *p* > 0.05. (**C**) Quantifications of non-tremor movements show no significant difference between *β1-AR^−/−^;β2-AR^−/−^;Car8^wdl/wdl^* mice at baseline or after being treated with propranolol. (**D**) Representative raw electrophysiological traces of cerebellar nuclei neuron activity in *β1-AR^−/−^;β2-AR^−/−^;Car8^wdl/wdl^* mice before (N = 2, n = 6) and after propranolol treatment (N = 2, n = 5). The line graph at the top for each condition represents the mean firing rate in Hz at each point in time for the 5 s spike traces shown below each line graph. Below the 5 s spike traces are magnified views of the spikes within the outlined boxes, spanning 500 ms. (**E**) Quantifications of *β1-AR^−/−^;β2-AR^−/−^;Car8^wdl/wdl^* cerebellar nuclei neuron activity before and after propranolol. Propranolol has no effect on the firing rate, mode ISI^−1^, CV, or CV2 in *β1-AR^−/−^;β2-AR^−/−^;Car8^wdl/wdl^* nuclei neurons.

## Data Availability

Data is contained within the article.

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
