# Peer review of "Propranolol Modulates Cerebellar Circuit Activity and Reduces Tremor"

_cells, 2022, doi:10.3390/cells11233889_

Round 1

Reviewer 1 Report

In this study, Zhou and colleagues examined the neural basis for propranolol, a β-adrenergic receptor blocker, as a treatment for essential tremor. Using Car8wdl mutant mice, an animal model that shows complex motor deficits including tremor, ataxia, and dystonia, they found that propranolol specifically alleviated the severity of tremor without affecting other motor performances. Because the cerebellum is a key locus in the tremor circuits, the authors subsequently investigated the propranolol effects on the cerebellar activity by performing in vivo electrophysiological recordings from Purkinje cells and cerebellar nuclei neurons. They discovered that propranolol reduced the firing rate and improved the firing regularity in both neuronal populations. Taking advantage of β-adrenergic receptor (β1 and β2) knockout mice, they further revealed that the effects of propranolol on tremor and cerebellar activity were largely abolished in the absence of these receptors, suggesting that β-adrenergic receptors in the cerebellum may mediate the therapeutic outcome of propranolol. Overall, the experiments are cleverly designed and carefully executed. The findings are novel and will have a great impact on clinical application of propranolol. The manuscript is very well written. The figures and tables are very nicely presented. I only have a few minor comments:

1. As Purkinje cells inhibit cerebellar nuclei neurons, the authors may discuss whether and how decreased Purkinje cell activity by propranolol (Figure 3) affects the firing patterns of cerebellar nuclei neurons (Figure 4). Would propranolol simply lower the intrinsic excitability of both types of neurons? A further explanation may help understand these observations.

2. It is convincing that the propranolol effects on cerebellar nuclei neurons are eliminated in the triple knockout mice (Figure 7). Would this be the case for Purkinje cells? The authors may consider presenting or discussing the effects of propranolol on Purkinje cells in the triple knockout mice. This reviewer does not advocate more experiments.

3. In Figure 7B, the tremor power in the triple knockout mice (baseline) appears smaller than that in the Car8wdl mice (Figure 1D). If this difference is significant, please explain why deletion of β1 and β2 receptors results in the tremor reduction.

4. In Figure S1, please indicate the signals representing β1 and β2 receptors that have disappeared in the double knockout mice. More images with a high resolution may help here.

5. It may help the readers if all the bar graphs are individually labeled with corresponding groups.

6. Line 354 (Results section), the phrase may be changed to “… all figures are listed in Supplementary Table 1.”.

Reviewer 2 Report

The manuscript by Zhou et al sheds light on the circuit mechanisms by which propranolol decreases tremor. Propranolol, which is a known beta-adrenergic receptor blocker, is a clinically used treatment for tremor. Given the emerging role for the cerebellum in tremor, the authors focus their study on the effect of propranolol on the cerebellar circuitry. As an experimental system, they employ the Car8wdl/wdl mouse model, which is known to exhibit ataxia and tremor. The authors show that propranolol reduces tremor in this mouse model without affecting other tested motor behaviours. They then go on to show using in vivo recordings that propranolol affects the firing activity of both Purkinje cells and deep cerebellar nuclei neurons and that propanol mediates its effect by acting on the beta-adrenergic receptors. Overall, this is an interesting study that contributes to our understanding of how propanol, a commonly used drug to treat tremor, might work mechanistically. The experimental design is solid, the figures are well-presented, and the manuscript is clearly written. I have no major concerns regarding this study but would like the authors to address a few points in order to strengthen the manuscript.

1. The authors conclude that propranolol selectively targets the cerebellar circuits responsible for tremor but not other motor behaviours, i.e., locomotor activity and gait. With regards to the locomotor activity, the authors use a previous dataset from a published study for the untreated mice. While this is mentioned in the materials and methods section, the authors should also clearly state this in Figure 2 and the results section.

From the footprint analysis of gait parameters, the authors conclude that propranolol has no effect on gait. However, Figure 2C suggests that propranolol does improve the step pattern of the mutant mice, as the front and hind paw prints appear to be more synchronized and regular after treatment with the drug. With the limited parameters that the authors assess in their manual footprint analysis, it cannot be ruled out that other, more subtle gait parameters might be improved.

Related to this, it is surprising that only the footprint test was utilized and no other motor behavioral assays, for example Rotarod testing, which is a well-established phenotype in the mutant mice and extensively used by the authors in their other publications on this mouse model. While I am not asking the authors to do additional extensive behavioural testing, it would be good to comment on the limited number of assays used.

2. Given that the characterization of the ataxic phenotype and the effects of propranolol on this is limited in this study, the authors should consider revaluating their claim that propanol targets tremor circuits specifically.

3. The expression of beta-adrenergic receptors was assessed in the cerebellar cortex (Figure 5) but not in the deep cerebellar nuclei or the inferior olive. However, functionally, the authors showed that propranolol treatment lowers complex spike firing rates and their ISI-I mode, suggesting that propranolol likely acts on the inferior olive. Moreover, direct action of propranolol on the DCN could explain the observed changes in firing activity. The authors should address this point and comment on the on the likely role of beta-adrenergic receptor’s role and function in the inferior olive and DCN.

4. Given the altered complex spike rates, the authors should add some discussion on how the complex spike changes could contribute to the changes in the Purkinje cell firing observed and alleviate the tremor phenotype.

Reviewer 3 Report

The authors utilized Car8 mice to study the therapeutic effect of propranolol in the cerebellar perspectives. Using tremor recording, open-field test and electrophysiology in-vivo, the authors suggested that the propranolol changes the cerebellar physiology, which contributes to the therapeutic mechanism of tremor. The authors provided comprehensive electrophysiological observations in both Car8 mice and beta1+beta2+Car8 triple KO mice. It is a pity that the authors did not provide evidence of cerebellar interventions to establish the causal relationship between cerebellar ephys changes and propranolol therapeutic effect. However, the correlation is consistent and worth publishing with some modifications of the content.  There are several key points and concerns in this paper:

Major point:

The major missing point of this manuscript is that there is no direct intervention in the cerebellum, such as intra-cerebellar microinfusion of propranolol or viral-mediated knock-down of B1/B2 receptors in CN. The therapeutic effects can come from extra-cerebellar mechanisms by systemic use of propranolol and global KO of beta-adrenergic receptors can alleviate propranolol therapeutics outside of the cerebellum. It is a pity that current observations in the cerebellum physiology cannot provide decisive evidence whether the cerebellar change is a predominant factor, one of the key factors, a minor factor or an epiphenomenon of the propranolol therapeutic mechanism. Due to this point, the authors should tune down their statement. For example, the last sentence of the abstract: “These data show that propranolol reduces tremor in mice by modulating cerebellar circuit activity through β-adrenergic receptors” may not be true, and should be tuned down to a statement such as  “Propranolol can modulate cerebellar circuit activity through β-adrenergic receptors and may contribute to tremor therapeutics”. Still, the evidence at the cerebellar perspective is valuable and worth publishing.

Minor points:

11.  For non-tremor related motions, the authors applied a small time-window (therefore with a poor frequency resolution) and analyzed the motion power at 50-100 Hz range. It should be noticed that with instrumental tremor recordings, harmonics of tremor frequencies are commonly seen and serves as an important confounder that may lead to different interpretation. If presented, the harmonics may extend to 50-100 Hz frequency range that may affect the “non-tremor” claim of the analysis. To exclude this possibility, the spectral power of tremor recordings should also cover the same 50-100 Hz and compared between propranolol (+) and (-) scenarios. Harmonics could be detected and excluded from the frequency ranges for “non-tremor” analysis. Also, the authors should provide raw instrumental sampling rate and filtering settings in the method section, to validate whether the analysis method is mathematically appropriate.

22.  For the footprint analysis of the data (Fig. 2C), the background gray (dashed?) lines are not mentioned in methods or legends, and extended beyond the graph boundary. I presumed that are the boundary of standard deviation. Please clarify the meaning of those lines. Also, their patterns are very different between controls with or without propranolol. Please also explain. 

33.  There is no electrophysiological difference in the triple-knockout mice in Fig. 7. It is possible to related to the N number of the data, which is only 6 data points in each group and the sample size 1/2-1/3 compared to previous experiments. For example, CV (Fig 7E) is likely to be reduced by propranolol, but lack of adequate statistical power. The N number is better to increase to approximate the same size. Due to the difficult breeding of triple-KO mice, the authors could repeat the same experiment to demonstrate result consistency.

Round 2

Reviewer 3 Report

The authors revised all issues addressed by the reviewers. I suggest to accept this manuscript in its current form.